

# Technical note: Effects of storage conditions on molecular-level composition of organic aerosol particles

Julian Resch[1], Kate Wolfer[1], Alexandre Barth[1], Markus Kalberer[1]

[1] Department of Environmental Sciences, University of Basel, Klingelbergstrasse 27, 4056 Basel, Switzerland

*Correspondence to*: Markus Kalberer (markus.kalberer@unibas.ch)

**Abstract.** A significant fraction of atmospheric aerosol particles, which affect both the Earth's climate and human health, can be attributed to organic compounds and especially secondary organic aerosol (SOA). To better understand the sources and processes generating organic aerosol particles, detailed chemical characterization is necessary, and particles are often collected onto filters and subsequently analyzed by liquid chromatography mass spectrometry (LC-MS). A downside of such

offline analysis techniques is the uncertainty regarding artefactual changes in composition occurring during sample collection, storage, extraction and analysis. The goal of this work was to characterize how storage conditions and storage time may affect the chemical composition of SOA generated from β-pinene and naphthalene, as well as from urban atmospheric aerosol samples. SOA samples were produced in the laboratory using an aerosol flow tube and collected on PTFE filters, whereas ambient samples were collected onto quartz filters with a high-volume air sampler. To characterize

temporal changes of SOA composition, all samples were extracted and analyzed immediately after collection, but were also stored as aqueous extracts or as filters for 24 hours and up to 4 weeks at three different temperatures of +20° C, -20° C or -80° C, to assess whether a lower storage temperature would be favorable. Analysis was conducted using ultra high-performance liquid chromatography high resolution mass spectrometry (UHPLC-HRMS). Both principal component analysis (PCA) and time series of selected compounds were analyzed to identify the compositional changes over time. We

illustrate that the chemical composition of organic aerosols remained stable during low temperature storage conditions, while storage at room temperature led to significant changes over time, even at short storage times of only one day. This indicates that it is necessary to freeze samples immediately after collection, and this requirement is especially important when automated ambient sampling devices are used where filters might be stored in the device for several days before being transferred to a laboratory.

## 25   1 Introduction

Organic aerosol (OA), and especially secondary organic aerosol (SOA), constitutes a large fraction of atmospheric fine particulate matter ($PM_{2.5}$) and has been shown to exert effects on both the climate and human health (Hallquist et al., 2009; Pöschl and Shiraiwa, 2015; Jimenez et al., 2009). The complexity of organic matter on the molecular level, representing thousands of different compounds, requires detailed and sensitive chemical characterization to identify the sources or





atmospheric processes generating the organic material (Johnston and Kerecman, 2019). Highly detailed chemical analysis can be hard to achieve with online measurement techniques (Stark et al., 2015; Nozière et al., 2015), and instead offline analysis (most commonly mass spectrometry) are necessary, where it is common that aerosol particles are collected on filters and analyzed at a later point in time in the laboratory.

Although offline methods enable very detailed chemical characterization and accurate quantification, they are prone to
multiple sample collection, work up and storage artifacts, which have the potential to alter the particle composition significantly, and thus confound the characterization of the original particle composition in the atmosphere. These influences have been discussed previously in the literature, including the use of different filter substrates, extraction methods and different storage times and conditions. Several studies (e.g., Parshintsev et al., 2011, Perrino et al., 2013) explored differences in aerosol composition between samples collected on quartz and PTFE membrane filters, and identified
significant gas phase adsorption artifacts, especially on quartz filters. These differences prevent the direct comparison of results between different studies, particularly where the filter materials used are not described. Other studies examined differences in extraction methods, with the notable observation that sonication causes $H_2O_2$ formation in aqueous extracts (e.g., Mark et al., 1998, Fuller et al., 2014). This is a particularly major problem for chemical characterization, as it triggers further reactions in the extracts, creating side products (which may themselves also be present in atmospheric particles), and
therefore leading to differences in results if not taken into consideration, while vortex extractions largely avoid such artifacts (Fuller et al., 2014). A study by Roper et al. (2019) compared different extraction methods of individual $PM_{2.5}$ filters, and observed significant differences in the concentrations of elements and polycyclic aromatic hydrocarbons (PAHs). More recently Wong et al. (2021) investigated the effects of water versus acetonitrile as extraction solvents on the chemical composition of SOA during storage for 1-2 days, and identified concentration changes for some components.

All of these studies exhibit how small differences in samples collection, extraction and storage can lead to different results and therefore highlight how important it is to characterize such potential artifacts in organic offline analysis measurements and carefully report sample workup conditions.

In addition, in multiple studies where aerosol particles are analyzed for their detailed organic composition, samples are stored on filters or as solvent extracts for a considerable amount of time, and analyses are sometimes performed months or
even years after the initial sample collection. The total storage time is often only indirectly or not at all recorded, which makes the assessment of the nature and extent of potential artifacts impossible. Extended storage on filters at room temperature may, for example, occur during automated sampling of high-volume samples. Storage conditions were often developed and evaluated for particle characterizations other than detailed organic molecular-level analysis, where extended storage has no significant effect (e.g. for total carbon, gravimetry, metal or inorganic ion analysis). However, if such filters
are also used for detailed organic compositional studies, then caution is needed to avoid unintended and unaccountable alteration of particle composition before analysis.

In this study we aimed to define the effects of different storage conditions and times on the molecular-level composition of organic aerosols using ultra high-performance liquid chromatography high resolution mass spectrometry (UHPLC-HRMS).



We characterized the changes occurring in organic aerosol particles collected on offline filter samples and stored as filters or as extracts at different temperatures from room temperature to -80° C, and for different time periods, from immediate analysis to four weeks storage time. We collected and characterized both laboratory-generated SOA particles and ambient atmospheric aerosol samples from an urban location.

## 2 Materials and Methods

### 2.1 Chemicals

β-Pinene, cis-pinonic acid, camphoric acid, 4-hydroxy benzoic acid, naphthalene, 1,2-naphthoquinone and pimelic acid were all obtained from Sigma Aldrich (Merck, Switzerland). Optima LC-MS grade water, methanol, acetonitrile, formic acid and acetic acid were obtained from Fisher Scientific (Switzerland). $PM_{2.5}$ ambient samples were collected on 150 mm PALL Tissuquartz membrane filters (VWR, Switzerland). SOA samples were collected on 47 mm PTFE membrane filters with 0.2 µm pore size (Whatman, Merck, Switzerland).

### 2.2 Filter sample collection and extraction

In this study laboratory-generated SOA and ambient samples were collected and characterized to cover a wide range of organic aerosol components.

Two precursors, β-pinene and naphthalene, representing natural and anthropogenic sources, were used to generate SOA particles via $O_3$ and OH oxidation with a compact aerosol flow tube, the "organic coating unit" (OCU) (Keller et al., 2022). The detailed setup for SOA generation, concentrations and masses deposited onto the filters are presented in the Supplementary Information (Fig. S1 and Table S1). Five filter samples were collected for each SOA type and storage condition to assess reproducibility. Prior to particle collection, each filter was cleaned, to remove residual organic products from manufacture by rinsing with LC-MS grade methanol, and air-dried in the fume hood.

Ambient $PM_{2.5}$ samples were collected with a Digitel DH-77 high-volume air sampler fitted with a $PM_{2.5}$ inlet (Digitel, Switzerland). The urban sampling site was on the roof of a building at 20 m height above street level at Klingelbergstrasse 27, Basel, Switzerland. Prior to sampling, each quartz filter was baked out for 6 hours at 550° C to remove residual organics and to ensure reproducibility; cleaned filters were stored at -80° C wrapped in aluminium foil and in an airtight plastic storage bag until use. High-volume ambient aerosol samples (HVAS) were collected at a flow rate of 500 L min$^{-1}$ for 24 hours. The exposure area of each filter was 169.7 cm$^2$.

An overview of all samples collected and the time between collection and extraction/analysis is given in Table S2. All samples were stored in the dark and at temperatures of either +20° C (hereafter referred to as room temperature), -20° C and -80° C, and were analyzed either immediately or after storage times up to 44 days. Due to the large number of samples and LC-MS analyses it was not possible to analyze all samples after the exact same number of days.



The filter extraction of SOA and ambient samples differed due to the difference in properties of the filter material, PTFE for SOA and quartz for ambient. The extraction is given below for SOA samples and deviations for ambient samples are indicated in parentheses.

Each filter was cut into equal quarters (for ambient filter samples five 1 cm punches were used), placed into 2 mL Eppendorf safe-lock tubes (Eppendorf, Switzerland) and placed in a freezer (i.e. -20° C or -80° C) or extracted immediately. For extraction 1.5 mL extraction solvent (1:5 water/acetonitrile (ACN) v/v) was added to the safe-lock tube and then the samples were vortexed at maximum speed (44000 rpm) for 2 minutes each and placed on a Fisherbrand™ Open Air Rocker (Fisher Scientific, Switzerland) for 30 minutes (post-extraction, ambient samples were additionally put in a centrifuge for 10 minutes at 12000 rpm to separate the quartz filter slurry from the liquid sample). 1.5 mL of the sample extract was then pipetted into an empty Eppendorf tube to remove the filter material (for ambient samples 1 mL of sample extract was transferred to an empty Eppendorf tube using a 5 mL gastight glass Hamilton syringe and a PTFE 0.45 µm pore size syringe filter (Agilent Technologies, Switzerland) to avoid larger particles from being transferred into the LC, a common source of blockages). The samples were then placed into a benchtop rotary evaporator (Eppendorf Basic Concentrator Plus; Eppendorf, Switzerland) and extracts were dried for 2 hours at 45° C in vacuum concentrator alcohol (V-AL) mode until complete dryness; this process was conducted in batches where necessary. Samples were then reconstituted with 500 µL (ambient samples in 400 µL, to further concentrate the samples) reconstitution solvent (1:10 ACN:water v/v) and vortexed again for 90 seconds before they were split into 5 aliquots of 100 µL (ambient samples: 80 µL) in amber LC-MS vials with 150 µL glass inserts. These were then either stored for the times stated in Table S2, or placed directly in the LC autosampler for analysis.

## 2.3 UHPLC-MS analysis

Liquid chromatographic measurements were conducted using a Thermo Vanquish Horizon UHPLC with binary pump and split sampler (Thermo Fisher Scientific, Reinach, Switzerland). For separations, a Waters HSS T3 UPLC column (100 mm x 2.1 mm, 1.8 µm, Waters AG, Baden, Switzerland) was used at a temperature of 40° C and a flow rate of 400 µL min$^{-1}$. Water + 10 mM acetic acid (mobile phase A) and methanol (mobile phase B) were used as mobile phases at the following gradient in a 30 minute method: 99.9% A from 0-2 min, a linear ramp up to 99.9% B from 2-26 min, 99.9% B was held until 28 min, then switching to 99.9% A for column re-equilibration from 28.1-30 min. To clean up between sample injections and prevent carryover, a needle wash using 1:4 ACN:H$_2$O (with 0.1% acetic acid) was performed for 15 s prior to each sample injection. Additionally, a seal wash of 1:10 methanol/water (with 0.1% formic acid) was used. To ensure system suitability, and stability of the signal intensities and retention times over multiple weeks of analyses, and for batch correction where necessary, an HPLC gradient test mix injection consisting of phenol, uracil and a mixture of parabens (Sigma Aldrich, Merck, Switzerland) was run daily.

An Orbitrap Q Exactive Plus (Thermo Fisher Scientific, Switzerland) was used for mass spectrometric detection in negative electrospray mode. Instrument parameters used were: spray voltage 3.5 kV, sheath gas flow 60 a.u., auxiliary gas flow 15, sweep gas flow 1, capillary temperature 275° C, auxiliary gas heater temperature 150° C. The scan parameters were set to



Full MS, scan range 85 to 1000 m/z, resolution 70000, and AGC target of 3E6 and a maximum injection time of 25 ms. The mass spectrometer was calibrated daily using Thermo Scientific Pierce Negative Ion Calibration Solution (Fisher Scientific, Switzerland). Additionally, a standard mix consisting of camphoric acid, cis-pinonic acid, 4-hydroxybenzoic acid, 1,2-

naphthoquinone and pimelic acid was run at concentrations between 10 ng mL$^{-1}$ and 0.01 mg mL$^{-1}$ to obtain calibration curves of compounds with atmospheric relevance and which were also used along with the HPLC gradient test mix to monitor the stability of signal intensity and retention times (see Sect. S2 and Table S3). Cis-pinonic acid and 1,2-naphthoquinone were additionally used for annotation.

In total 810 (270 per sample type) LC-MS injections were run, including repeats and excluding blanks and conditioning runs.

Raw data files were converted to mzML format using Proteowizard (MSConvert, Version 3) software (Chambers et al., 2012). LC-MS data analysis was performed in R 4.2.1 (R Core Team, Austria) in RStudio 2022.07.1 (Boston, MA, USA) using the XCMS package for untargeted peak detection (Smith et al., 2006; Tautenhahn et al., 2008; Benton et al., 2010) and the peakPantheR (Wolfer et al., 2021) package for targeted feature extraction. For the untargeted analyses, the XCMS centWave algorithm was used for peak detection on the centroided data, to produce a table of m/z-retention time (RT) pairs,

referred to henceforth as features. All reported features are assumed to be the deprotonated (i.e. singly charged, [M-H]$^{-}$) species unless otherwise indicated. Additional in-house scripts using R and Python were used for post-processing data analysis.

To observe variation and trends in the large datasets produced, principal component analysis was used, as this method easily illustrates the dominant sources of variation in multivariate data. Multivariate statistical analysis was performed with

SIMCA® 17 (Sartorius, Germany); model performance was evaluated using $R^2$ values as a measure of proportion of variance explained by the model, and also by the $Q^2$ value, which estimates the predictive power of the model through 7-fold cross-validation using randomly selected test/train subsets taken from the whole dataset. The Hotelling's $T^2$ statistic was used to estimate potential outlier samples in the PCA scores relative to the whole dataset, using the multivariate probability distribution. The ggplot2 package (Wickham, 2016) in R was used to plot the principal component analysis (PCA) scores

plots from the SIMCA data. Python (Van Rossum and Drake, 2009) implemented in Spyder IDE 5.1.5 (Raybaut, 2009) with the Matplotlib (Hunter, 2007) and NumPy (Harris et al., 2020) packages was used for time series plots.

Error bars in the time series plots using the peak area represent the total relative uncertainty of ± 20%. This was calculated as the sum of the following individual uncertainties: The standard deviation of UHPLC-MS injection repeats which was 4%, the standard deviation of the detected peak area for specific features of the filter sample repeats which was 13% and the

variation due to the filter extraction procedure which was calculated from the immediately extracted samples and which was as high as 23%.



## 3 Results and Discussion

The main focus of this study was to evaluate potential effects of storage conditions, i.e. time, temperature and storage on filter versus extract, on the concentration of organic aerosol components in laboratory-generated SOA and ambient urban

aerosol. The samples were analyzed with UHPLC-Orbitrap MS and peak areas of all detected peaks in each chromatogram were compared using multivariate statistical analysis to identify overall trends. In addition, the peak area of the most intense peaks in the base peak chromatogram (BPC) for each sample type were investigated in more detail.

### 3.1 Laboratory-generated SOA from β-pinene

β-pinene was chosen as a representative biogenic precursor for SOA (Hallquist et al., 2009). In order to reduce the large

number of total features detected and remove potential interferences from non-informative noise and background peaks, a peak intensity filter was set to 7E5, hence only features with a peak intensity higher than this value were considered for further analysis. This led to 4735 features being detected for each of the 270 β-pinene SOA samples analyzed (excluding blanks); this figure is comparable to previous studies, with a similar number of features being detected in ambient $PM_{2.5}$ samples using LC-MS characterization (Pereira et al., 2021).

The PCA scores plot of principal components (PC) 1 and 2 (Fig. 1) shows that for samples stored as extracts and as filters, the key parameter to ensuring stable sample composition over weeks was the storage temperature. The samples immediately extracted and analyzed on the day of collection represent the freshest samples available, and the tight clustering of these indicates the stability over time and reproducibility of the aerosol generation and extraction. Both frozen sample types demonstrated little deviation in the multivariate space from the fresh samples, which confirmed the initial assumption that

keeping both extracts and filters at cool temperatures best preserves the chemical profile for at least several weeks as represented by the peak intensity for SOA samples.

In contrast, samples kept at room temperature drift away in the PCA model from the fresh samples, indicating a change in composition. Samples stored as filters or extracts at room temperature displayed a different behavior in PC1 and PC2 (PC1 giving the biggest variance for the filters and PC2 for the extracts). This suggests that there is a significant difference

between samples which are extracted immediately and ones which are kept as filters at room temperature. For these room temperature samples, there is a clear temporal trend over the storage time of about four weeks: the longer the samples were kept at room temperature, the larger the deviation from the fresh samples (see also Fig. S2, displaying the storage time for each data point). Both the filters and extracts stored at room temperature for two and four weeks even unveil signals outside of the Hotelling's $T^2$ ellipse representing the 95% limit of the multivariate probability distribution for the dataset, indicating

that if the sample set was unknown these samples might be qualified as outliers. The filters and extracts stored at room temperature seem to change their overall composition most significantly during the first days of storage, because the biggest change per day seems to occur at the beginning of the storage time (Fig. S2).



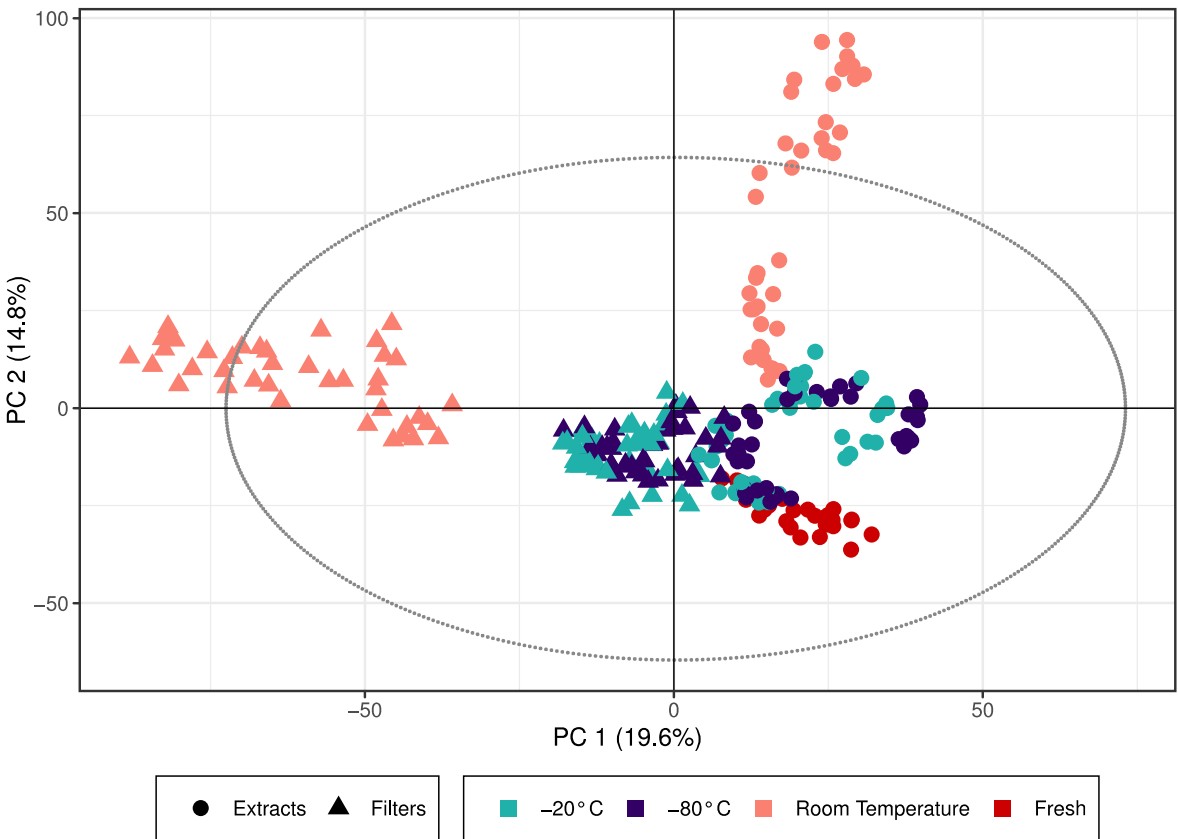

**Figure 1: PCA scores plot of the β-pinene SOA samples. Colors represent storage temperature and the directly analyzed (i.e. fresh)**
**samples and icons indicate the storage type. Hotelling's T2 ellipse (95%) represented by the dotted line. $R^2X[1]$ = 0.196, $R^2X[2]$ =**
**0.148, $Q^2[1]$ = 0.190, $Q^2[2]$ = 0.176.**

The four most intense peaks in the base peak chromatogram (see Fig. S3) of the immediate extracts were chosen as representative of how the relative concentration of individual chromatographic peaks change over time under the different storage conditions. Figure 2 illustrates these temporal trends, sorted by retention time, where each point represents the
average of two repeated analyses of each of the five filters collected (i.e., the average of 10 UHPLC-MS analyses). All four compounds or isomers of these have been identified in previous studies as carboxylic acids in SOA from gas-phase oxidation of α-pinene/β-pinene (Glasius et al., 2000; Yasmeen et al., 2010; Sato et al., 2016) and we tentatively confirmed the $M_w$ 184 (detected as m/z 183.1027, $C_{10}H_{15}O_3$,) peak at 11.74 min to be cis-pinonic acid through comparison with an authentic standard.

The time series plots show a similar trend to the PCA results: the samples kept at -20° or -80° C demonstrated the highest stability, where peak areas are also mostly within 25% of the values detected in the freshly analyzed samples for $M_w$ 172 (detected as m/z 171.0663, RT 6.73 min, $C_8H_{11}O_4$), $M_w$ 200 (detected as m/z 199.0976, RT 7.20 min, $C_{10}H_{15}O_4$) and $M_w$ 186



(detected as m/z 185.0819, RT 8.34 min, $C_9H_{13}O_4$). This clearly indicates that storing the samples at -20° C or below conserves samples sufficiently to prevent significant changes to these highest-intensity peaks.

In contrast, for features with $M_w$ 172, 186 and 200, the extracts and filters at room temperature demonstrated often pronounced increases over time (Fig. 2). This observation seems to contradict the hypothesis that compounds decay during storage. However, a possible explanation for this increase in these prominent features might be a decomposition of oligomers (i.e. compounds with 11 or more carbon atoms). Since it is assumed there is limited oxidation chemistry occurring during storage, it is unlikely that the concentration of these compounds increased due to oxidation reactions, which is the dominant

formation pathway of these compounds in the atmosphere. One class of oligomers frequently described in the literature are dimer esters (Hall and Johnston, 2012; Kenseth et al., 2018; Kristensen et al., 2016). The hydrolysis of dimer esters in samples stored in aqueous solution results in the precursor monomers as decomposition products (i.e. compounds with ten or fewer carbon atoms) (Zhao et al., 2018), which in our case would be the carboxylic acids discussed here. A time series analysis of a the dimer ester $M_w$ 388 (detected as m/z 387.0759, $C_{18}H_{28}O_9$) (Kristensen et al., 2016) is given in Fig. S4. This

compound showed a clear decrease over time for samples stored as extracts at room temperature, and it might therefore be one of the compounds decaying in the sample, causing the observed concentration increase of compounds presented in Fig. 2.

An exception to this trend is cis-pinonic acid ($M_w$ 184, RT 11.74 min), which had little temperature dependency, but the signal dropped by about 75% for the samples which were kept on the filters, whereas it remained relatively stable in

immediately extracted samples. Previous studies observed similar results, where cis-pinonic acid demonstrated different behavior in comparison to the rest of the dataset, i.e. with a desorptive loss upon purging spiked filters with clean air (Glasius et al., 2000) or a decrease in acetonitrile and an increase in water over time (Wong et al., 2021).

Overall, the results for β-pinene SOA demonstrate that samples, both extracts and filters, kept at temperatures of -20° C or below exhibited good stability of signal intensity over time, emphasizing that for studies conducting detailed offline analysis

of SOA composition samples should be immediately frozen after collection until analysis. However, these results also indicate that at least some compounds change over time, even under these low-temperature storage conditions, and the impacts of these artefacts on quantitative and compositional analyses must be considered. For samples kept at room temperature, there were clear and significant temporal changes of signal intensity for many features as illustrated in Fig. 1 and 2 and samples stored for a day or longer at room temperature before analysis should not be considered for detailed

chemical characterization.



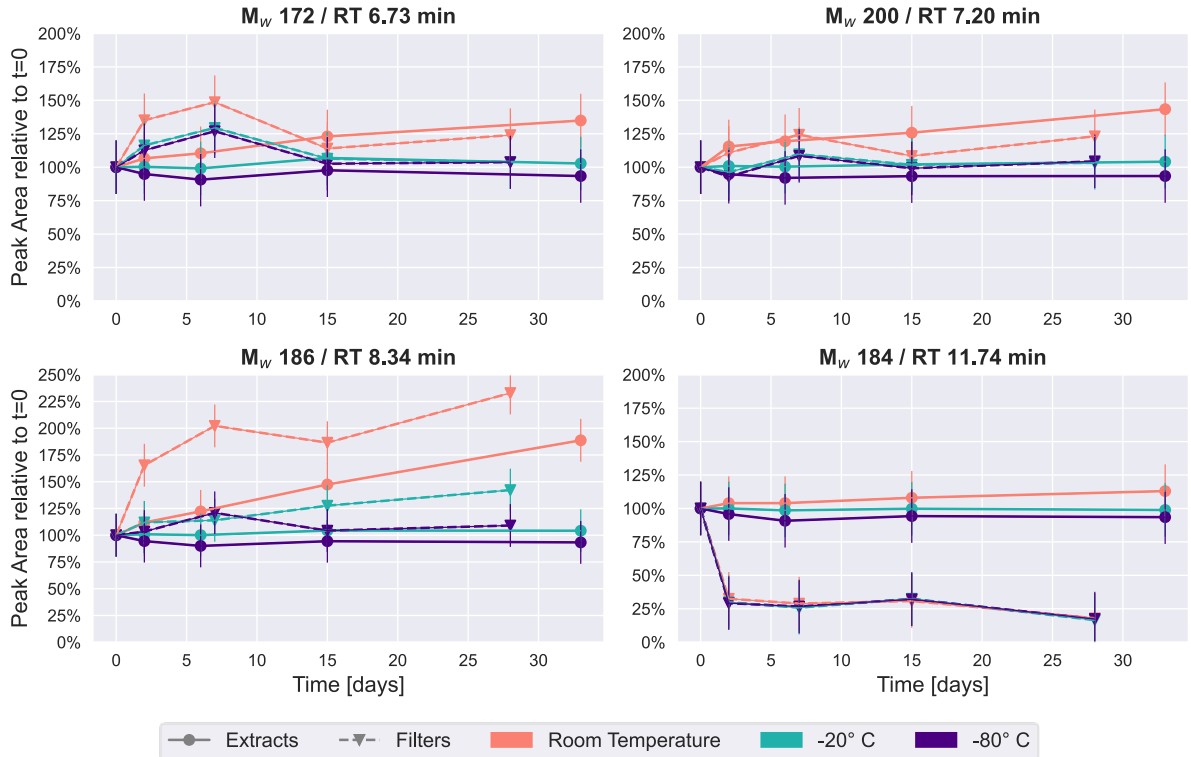

**Figure 2: Time series plots of the four most intense peaks in the β-pinene SOA samples over a period of 4-5 weeks. Especially for room temperature storage conditions, the concentration of some of these four compounds changes considerably.**

### 3.2 Laboratory-generated SOA from naphthalene

Naphthalene SOA (a representative anthropogenic aerosol (Eiguren-Fernandez et al., 2004)) samples were analyzed analogously to the β-pinene samples. A total of 5640 peaks with an intensity higher than 7E5 were detected in 269 analyses. The PCA scores plot for naphthalene-SOA in Fig. 3 displayed similar overall trends as for β-pinene-SOA (see Fig. 1). The generation of naphthalene SOA particles in the flow tube is slightly more unstable than for β-pinene, therefore the spread of fresh samples was higher across the five filter repeats as compared with the β-pinene samples. Similar to β-pinene SOA, the

naphthalene samples kept frozen at -20° C or at -80° C exhibited closer profiles to the immediately analyzed samples, and deviated little beyond the spread of the freshly extracted samples in the PCA model. For the room temperature samples there was a clear trend of variation associated with storage time for the extracts, which showed the largest variation in PC1, and for the filters, which showed largest variation in PC2. This similarity with the β-pinene samples indicates again that the overall composition of the SOA samples stored for 2-4 weeks at room temperature deviated significantly from the

immediately analyzed samples, and that the influence of extract and filter storage results in very different compositional changes. The samples kept at room temperature for 2-4 weeks fell outside the Hotelling's $T^2$ ellipse (see Fig. S5), again indicating that relative to the other samples, they have differing profiles and much larger variance across their features.



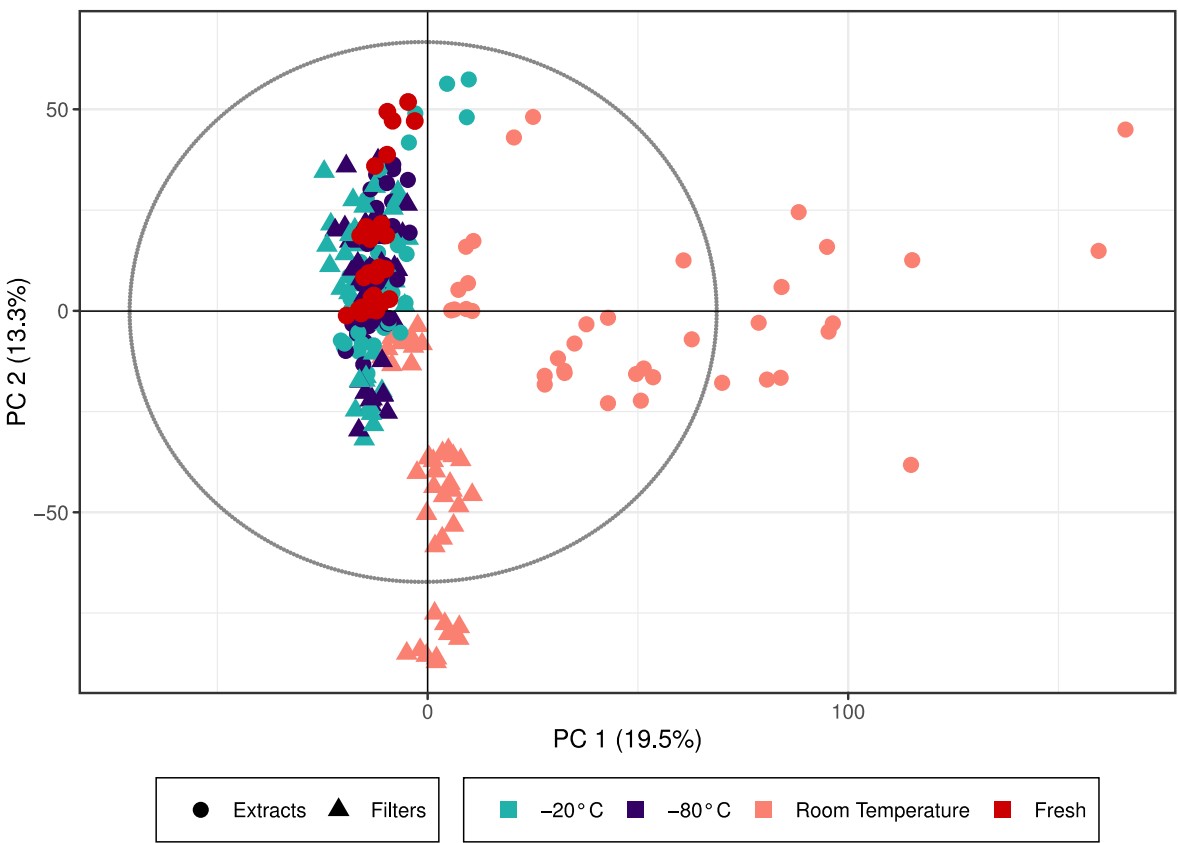

**Figure 3: PCA scores plot of the naphthalene-SOA samples. Colors represent storage temperature and the directly analyzed (i.e.
fresh) samples and icons indicate the storage type. Hotelling's T2 ellipse (95%) represented by the dotted line. $R^2X[1] = 0.195$,
$R^2X[2] = 0.133$, $Q^2[1] = 0.135$, $Q^2[2] = 0.153$.**

These trends were also visible for the four most intense peaks in the base peak chromatogram of naphthalene-SOA samples
as presented in Fig. 4. Again, the most stable storage conditions were freezing of the samples, and extracted samples
indicated a slightly improved temporal stability over the samples stored on filters in the freezers. The most noticeable
changes occurred for samples kept at room temperature. The most significant decay over time at room temperature was seen
for $M_w$ 158 (detected protonated anion of $M_w$ 158: m/z 159.0451, $C_{10}H_7O_2$) at 13.26 min, which was identified as 1,2-
naphthoquinone through comparison with an authentic standard. For extracts, the signal intensity dropped to less than half in
the first 24 hours, before disappearing completely in the samples analyzed after 1 – 4 weeks, and it appeared stable only
when stored at -80° C as the extract. 1,2-naphthoquinone is of increasing interest in the literature, as oxidized Polycyclic
aromatic hydrocarbons (PAHs) are known to cause oxidative stress in human lung cells, and thus are a direct contributor to
particle toxicity from anthropogenic sources (Kelly, 2003). It is evident from this data, that particle extraction and storage
conditions need to be carefully described and considered when these compounds are used for source apportionment, or to
infer particle health effects from laboratory-generated samples.





M$_w$ 166 (detected as m/z 165.0192, RT 6.83 min, C$_8$H$_5$O$_4$) has previously been found in naphthalene SOA samples and

identified as phthalic acid (Kleindienst et al., 2012). The most stable conditions for this compound were again observed when samples were kept frozen, while in extracts stored at room temperature, this compound steadily increased to almost double the intensity after a month. A possible explanation for the increase of especially M$_w$ 166 could again be the decay of oligomeric compounds causing an increase in their monomeric counterparts.

The other isomers of M$_w$ 210 (detected as m/z 209.0455, RT 5.72 min, C$_{10}$H$_9$O$_5$) and M$_w$ 150 (detected as m/z 149.0243, RT

7.50 min, C$_8$H$_5$O$_3$) selected for Fig. 4 showed moderate changes in comparison to the previously discussed compounds. Both exhibited relatively little change over time in samples which were kept in the freezers. The largest time-related effect can be seen for the samples kept at room temperature, where there is either a decrease (M$_w$ 210) or an increase (M$_w$ 150) of around 40% after 4 weeks.

These four most intense peaks contributed the most to the variance observed in the PCA scores plots, thus driving the

separation of samples by storage condition, and again reinforce the requirement to store organic aerosol samples in a freezer to best preserve their original composition.

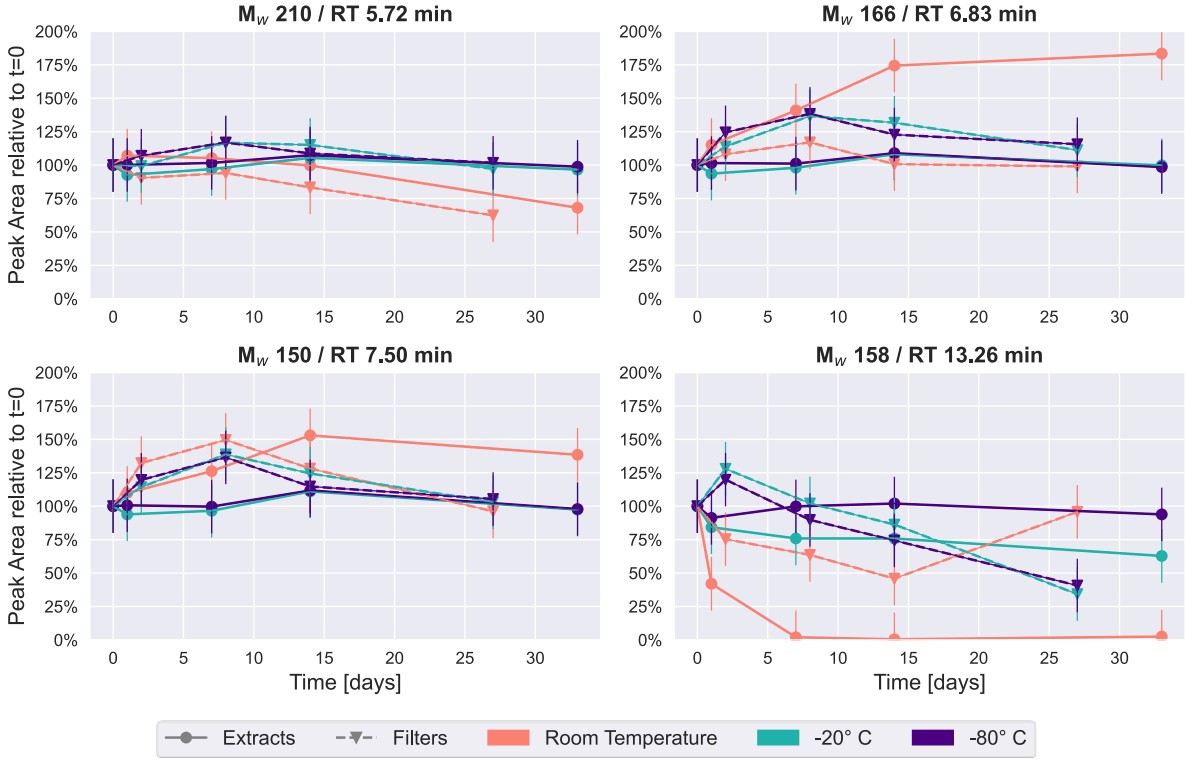

**Figure 4: Time series plots of the four most intense peaks in the naphthalene SOA samples. Similar as for b-pinene SOA (Fig. 2), room temperature storage significantly affects the concentration of some compounds in naphthalene SOA.**



**3.3 Atmospheric Aerosol**

To assess if the significant temporal trends and the differences in storage (i.e. filter vs. extracts) observed for laboratory-generated SOA samples were also visible in ambient samples, we collected five high-volume ambient aerosol samples in the city center of Basel and analyzed, extracted and stored them using the same methods as for the laboratory-generated SOA samples.

The PCA scores plot for the collected ambient samples is given in Fig. 5. The colors represent the five different HVAS samples and shapes correspond to storage temperatures. More detailed information on storage temperature and type are given in separate scores plot in the SI (Fig. S6 and S7). During LC-MS analysis of the ambient samples, a different batch of UHPLC grade water from the supplier was needed for the samples stored for 3-4 weeks, causing higher background signals and a reduced overall signal intensity for peaks with lower intensities. This difference in signal intensity could be adjusted for in the time series analysis of the compounds previously detected in SOA samples through the intensity of our standard mix, but was difficult to account for in the PCA. In order to solve this problem, the peak intensity parameter was increased from 7E5 (as used for the SOA samples discussed above) to 4E6 to reduce the number of total compounds detected from 2800 to around 400, because the higher intensity peaks were not significantly affected by this increased background. Additionally, a time series of the signal intensity of individual compounds was checked manually to exclude ones which had a clear "step-function", leaving roughly 240 compounds to be included in the PCA. The non-corrected version of the scores plot is given in Fig. S8, where the same general trend is still visible as in Fig. 5, S6 and S7.

In strong contrast to the laboratory-generates SOA samples, the PCA scores plots for the ambient samples indicated little storage-dependent variation in the signals, as samples grouped together in the first two PCs independently of storage temperature or condition, indicating a much larger influence of individual sample in the variance than from the storage condition. The HVAS samples from days 3-5 showed similar scores, as they were all sampled in the same week or even on consecutive days. To ensure that there was no additional variation between the storage temperatures we also looked at PCA scores plots of the individual HVAS samples, which presented the same trends (data not shown).

We conclude that in ambient samples the concentration of organic components is overall more stable over time and is apparently less affected by storage conditions compared to laboratory-generated SOA samples. This could be due to several factors. Organic components in ambient particles originate not only from SOA sources but include many primary particle components from other sources such as biomass burning, fossil fuel combustion, industrial activities (e.g. solvents) and primary biological material (Seinfeld and Pankow, 2003). Components from these sources might be more stable than SOA components. In addition, in ambient samples, a significant fraction of the total particle mass are inorganic components (mainly ions like sulfate, nitrate and ammonium) resulting in a more dilute concentration of individual organic components (compared to pure laboratory-generated SOA samples), which might limit the availability of organic reaction partners and thus increasing the stability of some organic components.




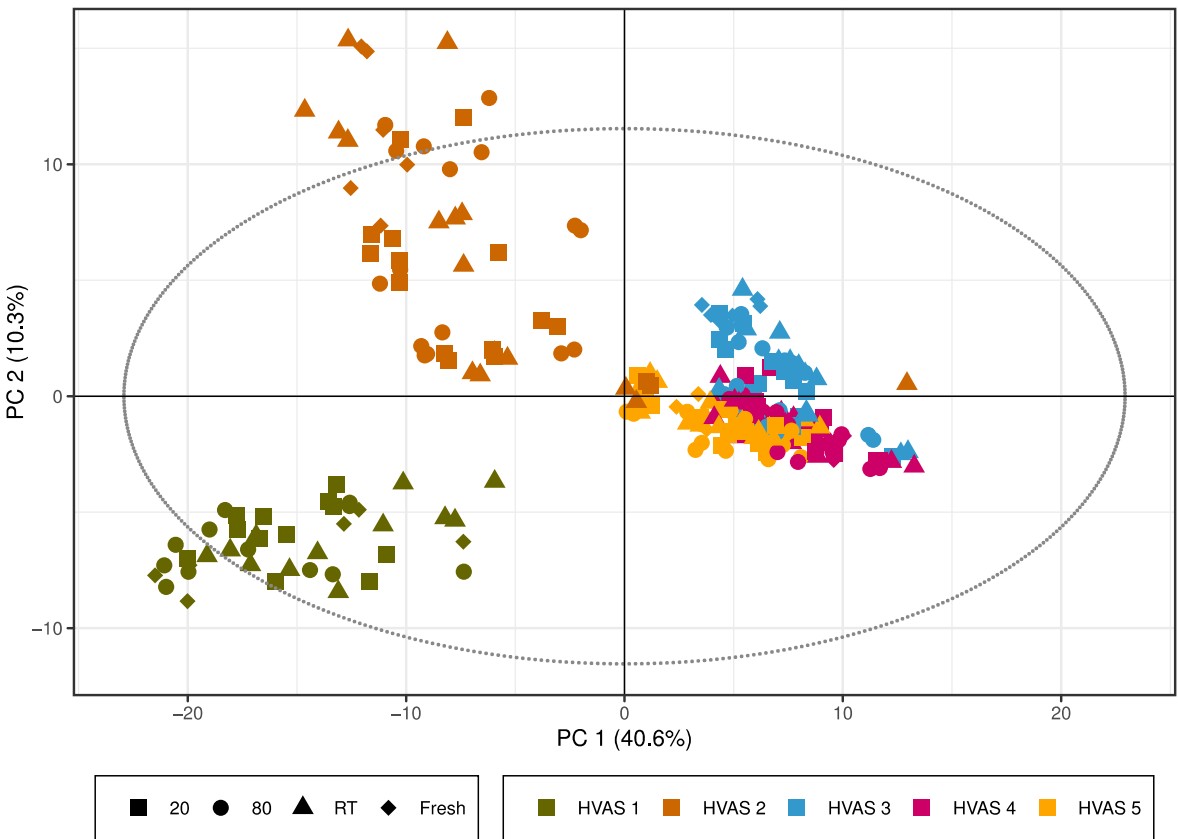

**Figure 5: PCA scores plot representing the HVAS filters with the exclusion of the batch effect due to different mobile phases. Colors represent the different HVAS filters and shapes the different storage temperatures $R^2X[1] = 0.406$, $R^2X[2] = 0.103$, $Q^2[1] = 0.399$, $Q^2[2] = 0.157$.**

For a compound-specific comparison between SOA and ambient samples, we analyzed four compounds which were detected in all HVAS samples, and which were also among the four highest peaks in the SOA samples (see Fig. 2 and Fig. 4). The time series for these compounds in the HVAS samples is given in Fig. 6. HVAS 1 was excluded from this analysis because of the missing 2-week time point (Table S2). Overall, these compounds were more stable over time in the ambient samples compared to the pure SOA samples, as also indicated in the PCA analysis, supporting the hypothesis that the lower concentrations of individual organic compounds in ambient aerosol leads to less signal change over time. This increased stability might also be due to the lower oligomer content in ambient aerosol in comparison to lab generated SOA (Kourtchev et al., 2016). Nevertheless, clear changes were observed for $M_w$ 166 and $M_w$ 186 for samples stored at room temperature and as extracts, which showed similar patterns to ambient and pure SOA samples. Slight changes over time (especially after 4 weeks) were seen for the $M_w$ 172 feature in the room temperature samples. The largest difference between ambient and laboratory SOA samples was observed for cis-pinonic acid ($M_w$ 184), where there was no significant difference between filter and extract storage in the ambient samples, but a large decay occurred in the pure SOA samples stored on filters.



Reasons for this very different behavior are unknown but could be related to the different filter material used for ambient and lab samples (quartz vs. PTFE). Another cause could be desorptive loss of cis-pinonic acid due to the large air masses in the

330 HVAS as previously reported (Glasius et al., 2000).

Overall, the storage of ambient samples on filters demonstrated very good stability of signal intensity, and provides confidence that the concentration of organic components may not change significantly in ambient urban samples which are collected weeks before analysis and which are stored on filters.

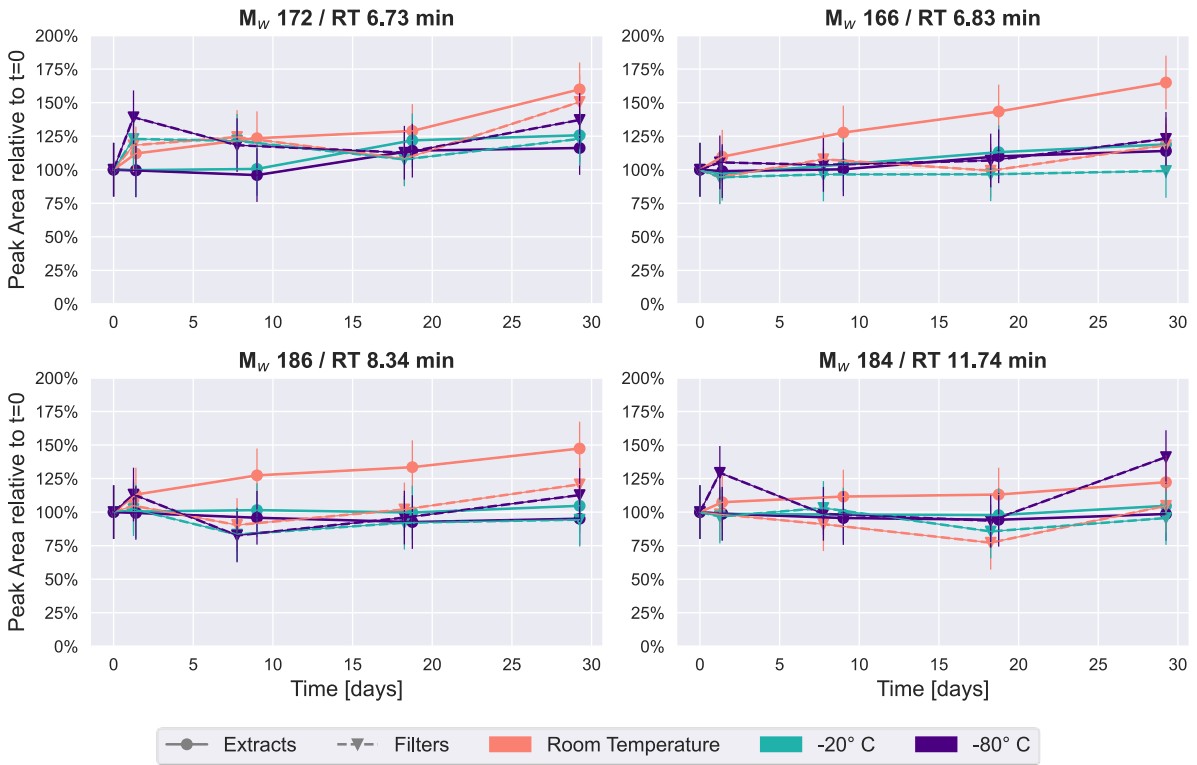

335 **Figure 6: Time series of an average of HVAS 2-5 samples of four previously detected peaks in the SOA samples from β-pinene (m/z 171.0663, m/z 185.0819 and m/z 183.1027) and from naphthalene (m/z 165.0192).**

## 4 Conclusions

The results in this study represent a thorough investigation of the temporal changes of the detailed organic composition of offline aerosol samples collected on filters under different storage conditions and for different types of aerosol. Both SOA

340 and ambient samples largely preserved their chemical profiles when stored at temperatures of -20° C or -80° C for up to 4-6 weeks. We could clearly demonstrate that there was no discernible difference in the particle composition when particles were stored at -20° C or at -80° C with the exception of very few individual components such as cis-pinonic acid (Fig.2).

However, for all investigated samples, but especially for lab-generated samples, storage of filters and of extracts at room temperature significantly affected the concentration of individual organic components, where compound formation as well as decomposition was observed. Many compounds with a high signal intensity in the chromatogram exhibited a significant increase in concentration over time when they were stored at room temperature. A possible explanation for this observation could be that some of these compounds are formed in the samples via decay of oligomers during storage, leading to the increase of their respective monomers. Keeping the samples frozen between collection and analysis appeared to largely avoid such decomposition reactions.

In many previous studies, the time between sampling and analysis is at least days, potentially up to many years, and often storage conditions are only poorly described in publications. The study presented here evidently indicates that careful storage procedures should be adopted and described in detail when reporting to assess potential distortions of the original particle composition, especially for laboratory or atmospheric simulation chamber samples, where significant changes can occur within a day after particle generation.

These compositional changes seemed to be less problematic for ambient particles at the urban site characterized here, but for some compounds concentration changes up to 50% or even more were observed also in ambient samples when analyzed several weeks after collection. Thus, when concentrations of individual organic particle components are studied in detail, a careful evaluation of their stability before analysis is demonstrably important, especially when samples are kept for days or weeks at room temperature, for example during automated filter sampling. In samples from other locations, e.g., remote sites, with higher or even dominant SOA contributions, the stability could be less favorable than for the urban samples analyzed here, and could resemble more the laboratory-generated SOA samples analyzed in this study.

**Author contribution**

JR, KW and MK designed the study. AB generated and collected the laboratory-generated SOA samples. JR collected the ambient aerosol samples and performed all other experimental and data analysis work. JR wrote the manuscript with contributions from all co-authors.

**Competing interests**

The authors declare that they have no conflict of interest.

**Acknowledgements**

This study was supported by the Swiss National Science Foundation (200021_192192/1).



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
