# Peer review of "Technical note: Effects of storage conditions on molecular-level composition of organic aerosol particles"

_EGUsphere, 2023_

## Author Comment (AC1)

**Referee 1:**

This study of filter stability before the analysis has been long overdue, and I am really happy the authors found the time to do it, and do it well! The conclusion is something people suspected but had no proof for: filters and extracts stored at room temperature change in composition significantly on a times scale of a day or so. This is important, and I expected many offline studies of aerosols to cite this study in the future. I recommend publishing after minor revisions.

I do not think this has to be be a technical note, it can also be a regular paper, especially if the authors can offer a hypothesis or a possible explanation for why aging processes in solution are different from those on a filter (Figures 1 and 3).

Although more work needs to be done to explain the different aging processes during storage on filters and in extracts, a possible explanation is now added to the conclusion section L 359-361.

Technical:

If it is practical I would make the X- and Y-scales on all PCA figures (Figures 1, 3, 5, S2, S5, S6, S7, S8) the same

In principle we agree that the same axis scales would make it easier to compare figures 1, 3, 5. However, because the y- and x-axis scales of these 3 figures vary by almost a factor of 10, this would compress a lot some of the data and especially figure 5 would be hard to read. Hence, we left the axis scales as they were.

Legend labels in Figures 5, S6, S7, S8 should say -20C and -80C instead of 20 and 80 (as they do in the rest of the PCA figures)

Thank you for noticing, all labels were changed in all these Figures.

Line 166 and line 236: Intensity of 7E5 is not too meaningful – it is better to specify it in relationship to the peak signal (such as X% of the largest observed peak). Similar comment for line 292.

We agree with this comment and added the percentage of the largest observed peak to lines 168, 240-241 and 300-301.

Figure 3: this figure can be much improved – the traces can be split into separate stacked panels, the text labels can be more readable, and the four most intense peaks mentioned in line 192 of the paper can be explicitly identified.

We assume the reviewer meant Figure S3. We separated the Chromatograms in Figure S3 (now Fig. S4) and added labels for the most intense peaks in the beta-pinene and naphthalene plot.

---

## Author Comment (AC2)

**Referee 2:**

This technical note presents the effect of the storage in different conditions on the molecular composition of aerosol produced in chamber and collected from ambient air. I absolutely agree with the authors on the crucial need to understand the effect of storage on the organic matter composition and I particularly appreciate the dual approach, taking into account the trends of specific m/z and on the global composition of the samples. The objectives of the study are clear and the methodology is well written and detailed. This work deserves to be published.

Nevertheless, few points can be improved:

The authors investigate 3 storage conditions for filters and extracts: room temperature, -20°C and -80°C. I would have added also 4°C. This could be extremely interesting for the extracts, because it is the temperature of the autosampler for many instruments (LC-MS, IC-MS…). Do the authors have an idea of the effect of storage in refrigerated conditions?

The reviewer mentions an important point and we agree that 4° C would be an important temperature to investigate because of the wide use of fridges and autosamplers. However, including another temperature condition would not have been feasible during this study but should be subject of future studies.

Unfortunately, it is not possible to predict the stability of organic particle components at 4° C from the data acquired here.

Regarding the storage conditions, at lines 56-61, the authors reported that storage conditions were evaluated for inorganic ions, EC and OC analysis. No references are reported. What are the current guidelines from research infrastructures? Are they comparable with those presented in this work?

We added several references discussing effects of storage on inorganic ions, EC and OC (Line 59-60). Because these studies do not compare storage conditions for molecular level organic particle composition (as we do it in our study), it is not possible to directly compare them with our work.

We are not aware of any research infrastructure guidelines.

SOA samples seems more perishable than ambient aerosol samples: few hours of sampling are enough to change the composition of the organic matter. Sampling in chamber should be performed in refrigerated conditions?

We did not look at changes in particle composition at 4°C and therefore cannot estimate how much this would slow down compositional changes.

For chamber experiments, it might be challenging to perform particle sampling at refrigerated conditions, but it would be advisable to collect particles for as short of a time as possible and freeze the samples immediately after collection.

The extraction procedure reported at line 98-99 is reported in previous studies or was designed for this study in particular?

This procedure was adapted from our previous study, Keller et al., 2022, and adapted for the needs of this study (mainly drying down samples in the concentrator instead of using nitrogen), as now explicitly stated in line 95-96.

The PCA enables a global view of the variation of the composition of the samples with storage. However, I miss some details of the statistic study, like the methodology for the normalization of the intensity of the peaks. This is essential when the analysis is performed on a large database like mass spectrometry results.

We thank the reviewer for pointing this out.

We re-analyzed the entire data set with $\log_{10}(x)$ normalized intensities. These results are shown in Fig. S3, S7, S9 and S11 and are very similar to the non-normalized data, the only difference between the normalized and the non-normalized PCA scores plots are the four week extracts in the β-pinene SOA. Corresponding text has been added on lines 178-180, 252-254 and 293.

I also feel that the PCA can be better exploited to understand the effect of the storage: is there a trend in the clusters?

We address this in lines 181-191. Labels in Figure S2 shows a temporal trend and clustering of the different storage times.

For example, looking at the room temperature extracts in Fig 1, I would expect that short storage times would be closer to the initial sample.

We were also surprised by this but it seems as if even a day of storage at room temperature is sufficient to change the chemical composition of samples significantly, both stored as filters and extracts.

I found very interesting the behaviour of cis-pinonic acid. May it be considered as a proxy to study the desorption or the sublimation of semivolatile organic compounds?

We agree, cis-pinonic acid could potentially be used in future studies as a proxy to detect and describe processes as the ones mentioned by the reviewer.

HVAS = ? high volume aerosol sample, I imagine

Yes, the abbreviation is given in line 88. HVAS = High-volume ambient aerosol samples.

I think that the objective of this study is to provide some guidelines on the storage of filters and extracts for aerosol samples. I expected to find these guidelines in the conclusions but I didn't. In my opinion, the authors should put the basis for a standardization of the methodologies in aerosol sampling for molecular characterisation and clearly state the procedure to adopt in order to improve the comparison of samples.

We agree with the reviewer. We added a sentence about a possible best practice for organic aerosol storage at the end of the conclusions section (line 375-378).

**References:**

Keller, A., Kalbermatter, D. M., Wolfer, K., Specht, P., Steigmeier, P., Resch, J., Kalberer, M., Hammer, T., and Vasilatou, K.: The organic coating unit, an all-in-one system for reproducible generation of secondary organic matter aerosol, Aerosol Sci. Technol., 56, 947–958, https://doi.org/10.1080/02786826.2022.2110448, 2022.

---

## Author Comment (AC3)

**Referee 3:**

L. 12: I suggest to include here which oxidants were used to generate the SOA.

This information is given in the main text and the authors aim to keep the abstract short and concise.

L20: It is not clear what you mean by "illustrate". Please use a more scientific description. Please add information about the temperatures.

"illustrate" has been changes to "show". The temperatures are listed in L16-17.

L21: I suggest to add more information about your results here to describe what you mean by "significant".

Details about the significant changes in the samples composition are given in the main text and we like to keep the abstract short and concise.

L53-61: This section contains speculations and broad criticism of previous studies. I suggest to change the wording to avoid these broad accusations of unspecified previous studies.

The authors think this comment is not a "broad accusation" but it is rather highlighting the fact that detailed information about sample storage conditions is often missing in previous studies.

L70: Please state purities if known.

The purities were added (L70-71).

Section 2.2: The conditions for laboratory generation of SOA samples should be described in much more detail including the OCU in the main manuscript. How much ageing does this compare to (oxidant concentrations, days)?

In L79 we refer to our previous publication where the OCU is described in much detail. Unfortunately, aging times or OH concentrations were not characterized in this study.

Furthermore the dates of ambient sampling should be listed, as well as supporting measurements such as PM levels.

The dates are listed in Table S2. No additional information was collected.

L90: Please provide this information, maybe in the SI.

L90 refers to Table S2 in the SI providing this information.

L101-102: What was the precision of these volumes? Please give the appropriate number of digits.

The type of Pipettes used and the extra digits were added to the text.

L113-115: These lines need revision regarding: liquid chromatographic measurements, split sampler, for separations. Please rewrite to clarify and improve the (scientific) language.

L115-116 was rewritten. "Split sampler" is the name given by the manufacturer.

Section 3.3. It is important to consider the influence of the time of year on the results.

All samples are collected in the same month (May 2022) as given in Table S2.

It is very interesting that the conclusion in section 3.3 is that ambient samples are less affected by the storage conditions. Could this difference be due to a high content of freshly formed reactive species (HOMs, peroxides etc.) in the laboratory-generated SOA?

Yes, this could be a reason.

Conclusion L359: This sentence seems quite speculative.

Our data shows that individual organic particle components change over time when kept at room temperature for ambient samples collected with a high-volume air sampler as discussed in section 3.3. Concluding that this needs to be considered for samples which are kept inside of an automated sampler at room temperature or above does not seem speculative.

Fig S4: Which type of SOA is shown?

This information was added to the caption (now Fig. S5).

Fig. S2 and S5: Numbers in dark blue markers are not visible.

The color of the labels was adjusted.

Line 10 artefactual – please check that this is a correct word

According to Collins dictionary it is.

L18: Seems that a word is missing after chromatography – with? coupled to?

No word is missing, this abbreviation is widely used in other studies and by the manufacturer.

L50: exhibit -> show?

Exhibit was changed to show.

L62: define – I assume you mean "identify" or "characterize".

Define was changed to identify.

L127: AGC target of 3E6 – please clarify

Automated Gain Control (AGC) was added to L129.

L164: I suggest to mention the oxidants again.

This information is given in the method section.

L297: generates -> generated

Generates was changed to generated.